# Surgery or Percutaneous Ablation for Liver Tumors? The Key Points Are: When, Where, and How Large

**DOI:** 10.3390/cancers17081344

**Published:** 2025-04-16

**Authors:** Paola Tombesi, Andrea Cutini, Francesca Di Vece, Valentina Grasso, Ugo Politti, Eleonora Capatti, Sergio Sartori

**Affiliations:** Section of Interventional Ultrasound, St. Anna Hospital, via Aldo Moro 8, I-44100 Ferrara, Italy; p.tombesi@ospfe.it (P.T.); a.cutini@ospfe.it (A.C.); f.divece@ospfe.it (F.D.V.); valentina.grasso@ospfe.it (V.G.); u.politti@ospfe.it (U.P.); e.capatti@ospfe.it (E.C.)

**Keywords:** liver tumor, liver resection, percutaneous thermal ablation, radiofrequency ablation, microwave ablation, contrast-enhanced ultrasound, fusion imaging, artificial intelligence

## Abstract

Recent comparisons between liver resection (LR) and percutaneous thermal ablation (PTA) reported similar survival outcomes for tumors ≤ 3 cm in size. Therefore, the debate should regard when LR or PTA are best suited to the individual patient. Subcapsular tumors or tumors closely adjacent to critical structures should undergo LR because ablation cannot achieve an adequate safety margin. PTA should be considered the first choice to treat central tumors (located ≤ 2 cm from the capsule) because it has lower complication rates, lower costs, and shorter hospital stays. Recent technical improvements in tumor targeting and assessment of the extent of the safety margin, such as stereotactic navigation, fusion imaging, and software powered by Artificial Intelligence, are changing the approach to tumors larger than 3 cm. The next trials should be aimed at investigating up to what tumor size PTA supported by these advanced technologies can achieve outcomes comparable to LR.

## 1. Introduction

In the first decade of this century, percutaneous radiofrequency ablation (RFA) was included with the curative treatment options for patients with early-stage hepatocellular carcinoma (HCC) who were not candidates for surgery [1]. After about ten years, the 2022 update of Barcelona Clinic Liver Cancer staging system (BCLC) recommended ablation as the first treatment approach for very early (BCLC-0) and early (BCLC-A) HCC patients who are not candidates for liver transplantation (LT) [2] because it is associated with survival outcomes similar to liver resection (LR) [3,4,5,6]. Likewise, the American Association for the Study of Liver Disease has recently recommended thermal ablation as an alternative option to surgery for patients with early HCC when LT is not feasible [7]. Furthermore, the final results of the COLLISION trial indicate that thermal ablation can even represent the standard of care for small liver metastases (LMs) from colorectal cancer [8,9]. This multicenter, international, randomized, controlled, phase III non-inferiority trial is the first one that compared ablation to LR for patients with ≤ 10 colorectal LM up to 3 cm in size. It showed that ablation offers comparable local tumor control and overall survival (OS) rates to those observed with LR, while being associated with a superior safety profile, shorter hospital stays, and lower costs [8,9]. However, despite these significant and increasing achievements, LR still remains the most popular treatment strategy worldwide, and percutaneous ablation is usually reserved to patients who are not surgical candidates [10,11,12].

## 2. Why?

### 2.1. The Past Weighs on the Present

An interesting editorial by Hsieh et al. [13] recently reviewed the role of LR and RFA in the treatment of early-stage HCC, comparing their pros and cons. The authors properly highlighted that the treatment landscape is evolving and requires a multidisciplinary approach, involving LT, LR, local regional therapies, and systemic treatments. In order to determine which approach is best for each single patient, a multidisciplinary team including surgeons, interventional radiologists, hepatologists, medical oncologists, and radiation oncologists should consider the patient’s liver function, functional status, comorbidities, cancer stage, and preferences [12,13]. LT is the sole treatment able to definitively cure both the tumor and underlying cirrhosis (when present), but it can infrequently be offered to patients because of limited donor availability and regulatory constraints [2]. When LT is not feasible, what is the optimal strategy for HCC patients with resectable tumors and good liver function? Despite the recommendations of the 2022 update of the BCLC staging system [2], patients classified as very early (0: single nodule ≤ 2 cm) and early-stage (A: single nodule or up to 3 nodules ≤ 3 cm) are often still considered ideal candidates for LR. Conversely, RFA is considered as an alternative when surgery is excluded, owing to concurrent morbidities or poor functional status, because it is safer and has a lower risk of complications than LR, but it yields worse outcomes and higher recurrence rates [13]. However, such an inferiority of percutaneous ablation compared to surgery is mostly based on many reports that appear somewhat dated. Two retrospective studies were published in 2008 and 2011 [14,15], and one study concluding that surgery was the gold standard treatment was published in 2020, but it retrospectively reviewed patients treated from 2007 to 2015 [16]. Moreover, a systematic review and meta-analysis published in 2020 concluded that LR is superior to RFA in promoting the survival of selected patients with resectable HCC, but it examined 17 studies (16 observational studies and just 1 randomized controlled trial) published from 2004 to 2011 [11]. Finally, a randomized controlled trial published in 2017 reported that RFA for early-stage HCC is not superior to LR in terms of tumor recurrence, OS, and disease-free survival, but it did not prove that RFA is inferior to LR [17]. In short, most (if not all) arguments supporting the superiority of surgery over percutaneous ablation in the treatment of small liver cancer do not take into account the progress in both ablation technology and imaging guidance systems that occurred in the last fifteen years. In all these papers, ablation was performed by using RFA, but last generation microwave ablation (MWA) systems were proved to achieve significantly larger ablation areas than RFA [18,19,20]. Currently, state of the art MWA devices equipped with advanced cooling systems and miniaturized quarter-wave impedance transformers to minimize the back-heating effect and to increase energy delivery into the target, can achieve single-antenna ablation sizes up to 5 cm, and possibly larger, in liver tissue [21]. Moreover, a propensity score-matched analysis for HCC ≤ 5 cm showed that MWA is comparable, if not superior, to LR in terms of tumor recurrence, disease-free survival, OS, and complications [22].

Likewise, imaging techniques aimed at guiding and monitoring the ablation procedure have seen significant progress. Ultrasound (US) remains the first-choice imaging modality to guide percutaneous ablation because it is cheap, easy to perform, does not require radiation exposure, and enables the real-time monitoring of each step of the ablation procedure [23]. Its main limitation, that is the poor visualization of the lesions because of small size, deep location, obscuration by overlying structures, or patient characteristics such as obesity or meteorism, has mostly been overcome by the implementation of low mechanical index contrast enhanced US (CEUS), and even more by the introduction into clinical practice of fusion imaging (FI) [13,21,24]. FI allows the fusion of real-time US or CEUS images with previously acquired computed tomography (CT) or magnetic resonance (MRI) volume datasets, enabling the precise localization and correct targeting of hardly visible or even not visible lesions, and visualization of tumors in high-risk locations [25]. Moreover, FI provides a better assessment of the ablation area and can precisely guide the immediate retreatment when residual viable foci are detected, dramatically improving safety and efficacy of percutaneous ablation [26,27,28]. Exploiting these technical improvements in both ablation technology and imaging guidance systems, the most recent comparisons between surgery and ablation report similar efficacy and survival outcomes for primary and secondary liver tumors ≤ 3 cm in size [5,6,8,9,29,30].

It is true that Mulier et al. [31], in 2005, stated that the short-term benefits of less invasiveness associated with the percutaneous ablation did not outweigh the longer-term higher risk of local recurrence, but it is very likely that, in 2024, they would no longer state it. In 1886, Patent Motorwagen, the first combustion engine car, was slower than a horse, but nowadays, no one would say anymore that horses run faster than cars!

### 2.2. Where We Are and Where We Should and Could Go

Since the most recent experiences highlight that percutaneous ablation and surgical resection are equally effective in treating small liver tumors in selected patients [5,6,8,9,29,30], the topic of discussion should no longer be what is the most effective treatment, but rather when (or, better, where) a treatment is best suited to the individual patient with small liver cancer and no contraindications to surgery. Many relevant tumor factors, such as microvascular invasion or presence of microfoci surrounding the tumor, play a critical role in risk recurrence and outcomes of both surgical resection and percutaneous ablation, and may influence the choice between the two treatment strategies. However, at present, their preoperative identification by conventional imaging technologies is poorly reliable, and radiomics models aimed at providing information on these issues are still under development and awaiting clinical validation.

Currently, in our opinion, single subcapsular tumors or tumors closely adjacent (≤1 cm) to critical structures or vulnerable organs such as major bile ducts and vessels (portal or hepatic vein), gallbladder, stomach, or bowel should undergo LR because ablation can often not achieve an adequate safety margin even using techniques such as hydrodissection and/or real-time temperature monitoring. Conversely, percutaneous ablation should be considered the first choice to treat central tumors, defined as tumors located at least 2 cm from the liver capsule, even in proximity of the liver dome [32,33]. Indeed, patients with a central tumor are usually not good candidates for surgical resection because of the risk of more injury to normal liver tissue and blood loss, and ablation offers similar survival outcomes with lower complication rates, lower costs, and shorter hospital stays [5,8,9,12,30]. Likewise, tumor location should also play a key role in decision making in the presence of up to three lesions up to 3 cm in size. In addition to the subcapsular or central location of the lesions, the number of the involved hepatic segments should also be taken into consideration. If the tumors are confined to one segment, LR may be a good option, but when more segments are involved, percutaneous ablation should be the first treatment option as it offers more normal tissue preservation, preventing liver decompensation, in particular, in the presence of liver cirrhosis. Indeed, in non-LT candidates with multifocal tumors, the 2022 BCLC update does not recommend resection but rather ablation for HCCs ≤ 3 cm and transarterial chemoembolization otherwise [2,34].

So far, in our opinion, this is state of the art. However, technical advancements in both surgical and ablative procedures are always in progress and the therapeutic landscape is in constant evolution, so we should aim at going beyond the current state of the art. Minimally invasive surgical approaches, such as laparoscopic and robotic surgery, reduce blood loss and wound pain, and combined with intraoperative ablation can be effective in treating surgically unfavorable tumors deeply located or multifocal tumors, maximizing the liver remnant compared to traditional LR [35]. Furthermore, important developments in imaging technologies are enabling the translation of preoperative images to intraoperative settings, with a significant impact on the success of liver surgery. Intraoperative imaging can optimize precision in LR and improve clinical outcomes for patients [36]. In particular, near-infrared (NIR) fluorescence imaging at a wavelength of 1000–1700 nm (NIR-II) is a very promising modality, as it can provide a high tumor-to-normal liver tissue signal ratio and an enhanced tumor detection rate [37,38]. Moreover, multi-modal imaging systems integrating NIR-II fluorescence imaging with traditional imaging techniques, such as CT, MRI, positron emission tomography (PET), and so on, can counterbalance its inherent limitations such as limited penetration depth and axial resolution [39]. Their implementation in clinical practice could play a key role in precisely detecting tumor size and boundary, ultimately improving LR success rate and reducing postoperative recurrences [39].

On the other hand, prior studies suggested that the so-called 3 cm barrier can be broken by percutaneous ablation, thanks to ongoing innovations in both imaging and ablation modalities [40,41]. Precise tumor targeting and ablation probe positioning are crucial to determine the success of percutaneous ablation, as well as the accurate assessment of the extent of the safety margin, which should ideally be at least 5 mm of normal tissue surrounding the tumor on all sides [20,42,43]. Stereotactic navigation techniques can dramatically enhance treatment precision and safety of thermal ablation, enabling defining trajectories that avoid injury to critical anatomical structures, and to quantify the accuracy with which ablation probes are positioned, by measuring the error between the planned “optimal” and the final “real” probe position [44]. Using stereotactic RFA for very large (≥8 cm) primary and metastatic liver tumors, Schullian et al. achieved 5-year OS rates of over 60% and over 20%, respectively [45]. However, the cut-off of 3 cm (or, in one study, of 5 cm) is still reported to have a significant influence on treatment outcome [46,47,48,49].

Indeed, local tumor progression (LTP) (and ultimately overall outcome of thermal ablation) strongly depends on the ability to obtain a concentric ablative margin of at least 5 mm, and tumor size plays a crucial role: as it increases, its spatial shape becomes more complex, making it more difficult to obtain adequate ablation margins [43,50,51]. Moreover, multiple needle insertions are often needed to cover the entire volume of the tumor. US/CT or US/MRI FI can help in the assessment of the ablation area, improving the evaluation of ablation completion and guiding the immediate retreatment of residual tumor foci or insufficient ablation margins [26,52]. Combining FI guidance and stereotactic MWA, Yang et al. [50] were able to better define the profile of both the tumor and surrounding structures, and to guide optimal needle insertion, achieving a technical success rate of 97.1% in the treatment of primary and secondary liver tumors 3–7 cm in size. However, achieving perfect synchronization between US and CT or MRI is crucial to ensure the proper functioning of real-time FI, and synchronization requires very skilled operators, and is complex and time-consuming. Recent algorithms powered by Artificial Intelligence (AI) can automate the synchronization process, ensuring precise alignment and registration of imaging data and enabling the lesion to be targeted more accurately, with improvement of the overall accuracy of the ablative procedure [53]. To further make synchronization easier, faster, and more accurate, new tools are being developed that are able to capture photographs of the skin surface and to compare them pixel-by-pixel with the volumetric acquisitions of CT or MRI (AutoSync^®^, Esaote S.p.A., Genova, Italy). In addition to the ease and speed of execution, the main advantage of this system is the ability to allow synchronization independently of patient positioning [54].

Furthermore, the side-by-side comparison between pre-ablation CT or MRI and post-ablation contrast-enhanced cross-sectional images plays a key role in the assessment of the completeness of the treatment and the size of the safety margin, as well as in guiding further needle insertions in the case of incomplete ablation [26,50,52,55]. Until recently, the comparison was based on visual inspection by clinicians, and it was exposed to the risk of errors due to breath movements, changes in patient’s position, and tissue biomechanical changes after ablation [56]. Nowadays, the non-rigid registration of pre- and post-procedure CT or MRI are considered mandatory for the precise assessment of the technical success of percutaneous ablation [57,58]. Quite recently, a new technology powered by AI was developed that integrates a pre-ablation and post-ablation control software directly into the US machine (Augmented Ablation Suite, AuAS^®^; Esaote, Genova, Italy). After an elastic registration of pre- and post-ablation CT datasets, it enables the immediate comparison between the pre-procedure planned margins and the ablation area, quantifying any residual viable tumor tissue or insufficient safety margin [54]. It is very likely that this technology will substantially improve the outcomes of percutaneous ablation in the near future, allowing for accurately targeting and successfully treating tumors larger than 3 cm in size.

However, these advanced imaging and navigation technologies aimed at improving both surgical and ablative approaches are quite recent, and their use is currently limited to selected centers dedicated to their definitive validation before their widespread implementation in clinical practice.

## 3. Conclusions and Future Directions

In an editorial published in the Journal of Surgical Oncology thirteen years ago, Burak et al. observed the following: “as thermal ablation technologies improve, it will be imperative to constantly re-evaluate the relative roles of resection and ablation for HCC” [59]. The first documented successful liver resection was performed in 1888 [60], and modern liver surgery began in the late 1950s with the study of the segmental anatomy of the liver and the introduction of the concept of anatomical resection [61]. In contrast, the first application of RFA for the treatment of liver tumors in humans was reported in 1993 [62]. In a nutshell, thermal ablation had a decades-long gap in evidence generation and development compared to surgical resection. In our opinion, such a gap has largely been bridged, and the current evidence highlights that percutaneous ablation and LR are equally effective in treating liver tumors ≤ 3 cm [5,6,8,9,29,30].

In patients with resectable liver cancer who are not candidates for LT, the next trials should be aimed at investigating up to what tumor size percutaneous ablation, supported by the most advanced ablation technologies and imaging systems to plan and guide the treatment and assess its completeness, can achieve outcomes comparable to surgical resection.

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
