# Peer review of "Surgery or Percutaneous Ablation for Liver Tumors? The Key Points Are: When, Where, and How Large"

_cancers, 2025, doi:10.3390/cancers17081344_

Round 1

Reviewer 1 Report

Comments and Suggestions for Authors

The authors demonstrated the evolving role of liver resection (LR) and percutaneous thermal ablation (PTA) in treating liver tumors, particularly those ≤3 cm. They present a well-structured discussion on when and where each treatment modality is best suited rather than debating absolute superiority. Furthermore, the manuscript highlights technological advancements in tumor targeting, fusion imaging, and artificial intelligence (AI)-based software, proposing PTA as a potentially viable alternative for tumors larger than 3 cm.

However, given that this is a review, some aspects need further refinement to position the manuscript as a forward-looking guide for future research rather than an argument for definitive clinical adoption of PTA. Below, I provide key recommendations to enhance the review’s impact and clarify its role in shaping future studies.

1. Clearly position the manuscript as a review article that summarizes current knowledge and identifies future research directions.
It is recommended that the introduction and conclusion explicitly state that this is a review article aimed at synthesizing existing evidence, outlining unresolved clinical and technological challenges, and proposing directions for future investigation.

2. Explicitly include a section discussing future research priorities and appropriate study designs.
For example, in light of the proposed extension of PTA indications beyond 3 cm tumors, the authors should suggest what types of prospective studies or randomized controlled trials are needed, including relevant outcome measures such as overall survival, recurrence-free survival, local tumor progression, quality of life, and cost-effectiveness. Adding such a section would significantly enhance the clinical and academic value of this review.

3. Clarify that AI-based imaging and navigation technologies are still in the validation phase before clinical adoption.
While these technologies may enhance the precision and efficacy of PTA, their current use is largely limited to selected centers or early-phase studies. The manuscript should strike an appropriate balance by clearly stating that further clinical validation is needed before widespread implementation.

4. Include a more in-depth discussion of tumor biology and its role in treatment selection between PTA and LR.
While advances in image guidance improve local control, factors such as microvascular invasion, tumor differentiation, and multiplicity continue to play critical roles in recurrence risk and long-term outcomes. The manuscript might have to address how these biological features may influence the choice between PTA and surgical resection.

5. Avoid overstating the equivalence between PTA and LR, and ensure consistent terminology throughout the manuscript.
Given that robust evidence from high-quality RCTs is still lacking, definitive statements suggesting equal efficacy may be premature. It is recommended to adopt more cautious language such as “PTA may offer comparable outcomes to LR in selected patients” or “PTA has the potential to match LR in certain clinical scenarios.” Consistent terminology (e.g., PTA vs. RFA vs. MWA) should also be maintained throughout the manuscript for clarity.

6. Minor comment

In simple summary, the definition of "central tumor" should be described as shown in main text. 

Author Response

1. Clearly position the manuscript as a review article that summarizes current knowledge and identifies future research directions.
It is recommended that the introduction and conclusion explicitly state that this is a review article aimed at synthesizing existing evidence, outlining unresolved clinical and technological challenges, and proposing directions for future investigation.
2. Explicitly include a section discussing future research priorities and appropriate study designs.
For example, in light of the proposed extension of PTA indications beyond 3 cm tumors, the authors should suggest what types of prospective studies or randomized controlled trials are needed, including relevant outcome measures such as overall survival, recurrence-free survival, local tumor progression, quality of life, and cost-effectiveness. Adding such a section would significantly enhance the clinical and academic value of this review.
3. Clarify that AI-based imaging and navigation technologies are still in the validation phase before clinical adoption.
While these technologies may enhance the precision and efficacy of PTA, their current use is largely limited to selected centers or early-phase studies. The manuscript should strike an appropriate balance by clearly stating that further clinical validation is needed before widespread implementation.
4. Include a more in-depth discussion of tumor biology and its role in treatment selection between PTA and LR.
While advances in image guidance improve local control, factors such as microvascular invasion, tumor differentiation, and multiplicity continue to play critical roles in recurrence risk and long-term outcomes. The manuscript might have to address how these biological features may influence the choice between PTA and surgical resection.
5. Avoid overstating the equivalence between PTA and LR, and ensure consistent terminology throughout the manuscript.
Given that robust evidence from high-quality RCTs is still lacking, definitive statements suggesting equal efficacy may be premature. It is recommended to adopt more cautious language such as “PTA may offer comparable outcomes to LR in selected patients” or “PTA has the potential to match LR in certain clinical scenarios.” Consistent terminology (e.g., PTA vs. RFA vs. MWA) should also be maintained throughout the manuscript for clarity.
6. Minor comment
In simple summary, the definition of "central tumor" should be described as shown

Response to Comments 1 and 2: sorry, but I’m afraid there is a misunderstanding: our manuscript is a Commentary, not a Review. Therefore, Comments 1 and 2, that would be totally on point for a Review, in our opinion are not suitable for a Commentary, that simply expresses our point of view based on the most recent developments of surgery and ablation technologies.

Response to Comment 3: I agree, and this issue has been highlighted in yellow in page 6, lines 7-10

Response to Comment 4: I only partly agree. It is true that biological features may influence the choice between ablation and surgical resection, but it is also true that we currently don’t have reliable tools to pre-operatively identify most of them, such as microvascular invasion or presence of microfoci surrounding the tumor. This observation has been added to the text and highlighted in yellow in page 3, lines 5-10 of section 2.2.

Response to Comment 5: : I only partly agree. “in selected patients” has been added to the statement that “percutaneous ablation and surgical resection are equally effective in treating small liver tumors” (page 3, line 2 of section 2.2). Conversely, I prefer to maintain the distinction between percutaneous ablation (PTA, that includes ablation techniques) and the single techniques (RFA or MWA), that are named when appropriate.

Response to Comment 6: I agree. Central tumor has been better defined (lines 5-6 of Simple Summary)

Reviewer 2 Report

Comments and Suggestions for Authors

This paper of commentary was well-written about the issue of Surgery or Percutanbeous ablation for liver tumors? The key points are, When Where  and How Large.

This issue has gradually been revealed to make clear evidence by the recent many trials.

Authors seems to be standpoints to ablation side relatively. However, this manuscript is treated as the commentary Therefore it could be accepted for the publication.

Author Response

Comment: Authors seems to be standpoints to ablation side relatively. However, this manuscript is treated as the commentary Therefore it could be accepted for the publication.
Response: actually, we hope that ablation can be considered with equal dignity to surgery!

Reviewer 3 Report

Comments and Suggestions for Authors

In this Commentary, Sergio Sartori et al. discuss the key considerations in selecting surgery or percutaneous ablation for liver tumors. They provide a comprehensive analysis of the current landscape of these treatment modalities.

Given that existing evidence suggests percutaneous ablation and liver resection are equally effective for liver tumors ≤ 3 cm and that percutaneous ablation may also be applicable to larger tumors, precise tumor size and boundary assessment is crucial for selecting the appropriate treatment strategy. Therefore, the authors should also discuss the role of advanced imaging techniques, such as near-infrared fluorescence imaging and multimodal imaging technology, in detecting liver tumor size and boundary, in addition to conventional modalities like US, MRI, and CT. A discussion on how these novel imaging approaches could influence the decision-making process for surgery versus percutaneous ablation would enhance the completeness of their analysis.

Additionally, there is an error in the reference numbering, as two references are assigned the serial number 1. The authors should revise the reference list to ensure correct numerical sequencing.

Author Response

Comment: Given that existing evidence suggests percutaneous ablation and liver resection are equally effective for liver tumors ≤ 3 cm and that percutaneous ablation may also be applicable to larger tumors, precise tumor size and boundary assessment is crucial for selecting the appropriate treatment strategy. Therefore, the authors should also discuss the role of advanced imaging techniques, such as near-infrared fluorescence imaging and multimodal imaging technology, in detecting liver tumor size and boundary, in addition to conventional modalities like US, MRI, and CT. A discussion on how these novel imaging approaches could influence the decision-making process for surgery versus percutaneous ablation would enhance the completeness of their analysis.
Additionally, there is an error in the reference numbering, as two references are assigned the serial number 1. The authors should revise the reference list to ensure correct numerical sequencing.

Response: 

Thank you very much for your suggestion. I have added a brief discussion on the role of near-infrared fluorescence imaging and multimodal imaging technology in improving the outcome of liver surgery in page 4, lines from 10 to 20 (marked in red), as well as four additional references (36-39) (marked in yellow).

Additionally, references have been corrected

Reviewer 4 Report

Comments and Suggestions for Authors

The authors discuss whether resection or ablation is more appropriate for individual cases. The authors note that both surgery and ablation are constantly making technical advances, and future clinical trials should aim to investigate the results of ablation versus surgery for tumors larger than 3 cm.

Overall well described.

Minor

There are no references 58

Author Response

Comment: There are no references 58
Response: thank you very much for your observation. References have been corrected (highlighted in yellow)